# Molecular evidence of SARS-CoV-2 in New York before the first pandemic wave

Matthew M. Hernandez[1,2,12], Ana S. Gonzalez-Reiche [3,12], Hala Alshammary[1], Shelcie Fabre[4], Zenab Khan[3], Adriana van De Guchte[3], Ajay Obla[3], Ethan Ellis[3,5], Mitchell J. Sullivan [3], Jessica Tan [1,6], Bremy Alburquerque [3,6], Juan Soto[3,5], Ching-Yi Wang[4], Shwetha Hara Sridhar[3,5], Ying-Chih Wang[3,5], Melissa Smith[3,5], Robert Sebra[3,5,7,8], Alberto E. Paniz-Mondolfi [2,4], Melissa R. Gitman[2,4], Michael D. Nowak[2,4], Carlos Cordon-Cardo [2], Marta Luksza[3,9], Florian Krammer [1], Harm van Bakel [3,5,13✉], Viviana Simon [1,10,11,13✉] & Emilia Mia Sordillo [2,4,13✉]

Numerous reports document the spread of SARS-CoV-2, but there is limited information on its introduction before the identification of a local case. This may lead to incorrect assumptions when modeling viral origins and transmission. Here, we utilize a sample pooling strategy to screen for previously undetected SARS-CoV-2 in de-identified, respiratory pathogen-negative nasopharyngeal specimens from 3,040 patients across the Mount Sinai Health System in New York. The patients had been previously evaluated for respiratory symptoms or influenza-like illness during the first 10 weeks of 2020. We identify SARS-CoV-2 RNA from specimens collected as early as 25 January 2020, and complete SARS-CoV-2 genome sequences from multiple pools of samples collected between late February and early March, documenting an increase prior to the later surge. Our results provide evidence of sporadic SARS-CoV-2 infections a full month before both the first officially documented case and emergence of New York as a COVID-19 epicenter in March 2020.

---

[1] Department of Microbiology, Icahn School of Medicine at Mount Sinai, New York, NY, USA. [2] Department of Pathology, Molecular, and Cell-Based Medicine, Icahn School of Medicine at Mount Sinai, New York, NY, USA. [3] Department of Genetics and Genomic Sciences, Icahn School of Medicine at Mount Sinai, New York, NY, USA. [4] Clinical Microbiology Laboratory, Department of Pathology, Molecular, and Cell-Based Medicine, Icahn School of Medicine at Mount Sinai, New York, NY, USA. [5] Icahn Institute for Data Science and Genomic Technology, Icahn School of Medicine at Mount Sinai, New York, NY, USA. [6] The Graduate School of Biomedical Sciences, Icahn School of Medicine at Mount Sinai, New York, NY, USA. [7] Black Family Stem Cell Institute, Icahn School of Medicine at Mount Sinai, New York, NY, USA. [8] Sema4, a Mount Sinai venture, Stamford, CT, USA. [9] Department of Oncological Sciences, Icahn School of Medicine at Mount Sinai, New York, NY, USA. [10] Division of Infectious Diseases, Department of Medicine, Icahn School of Medicine at Mount Sinai, New York, NY, USA. [11] The Global Health and Emerging Pathogens Institute, Icahn School of Medicine at Mount Sinai, New York, NY, USA. [12] These authors contributed equally: Matthew M. Hernandez, Ana S. Gonzalez-Reiche. [13] These authors jointly supervised this work: Harm van Bakel, Viviana Simon, Emilia Mia Sordillo. ✉email: harm.vanbakel@mssm.edu; viviana.simon@mssm.edu; emilia.sordillo@mountsinai.org

The first cases of coronavirus disease 19 (COVID-19), caused by severe acute respiratory syndrome coronavirus 2 (SARS-CoV-2), were observed in China in December 2019[1,2]. Within weeks, cases were reported in other countries in Asia, as well as in Europe and North America. In the United States (US), the first SARS-CoV-2 infection was confirmed by the US Centers for Disease Control and Prevention (CDC) on 20 January 2020[3]. During the following weeks, sporadic cases were reported throughout the US. When the first case in New York State (NYS) was diagnosed in New York City (NYC) on 29 February 2020[4], the NYC metropolitan area quickly emerged as an early epicenter of the pandemic.

We previously documented multiple independent introductions of SARS-CoV-2 into the NYC metropolitan area based on SARS-CoV-2 genomes obtained from 84 patients with COVID-19 receiving care at acute care hospitals and affiliated outpatient facilities of the Mount Sinai Health System (MSHS) during March 2020[4]. Based on phylogenetic reconstructions, we and others estimated that these independent SARS-CoV-2 introductions had occurred no later than early February 2020, and potentially as early as 8 January 2020[4,5]; this timeframe is further supported by our recent cross-sectional serosurvey of MSHS patients[6]. However, prior to mid-March, 2020, COVID-19 case detection in New York was limited by restricted availability of diagnostic testing and overlap in symptom presentation with other respiratory and viral illnesses. Thus, direct molecular evidence of SARS-CoV-2 in NYC prior to the first reported case has been lacking. Accurate information regarding these earliest infections is essential to understand and model virus transmission and to assess interventions aimed at controlling the spread. Herein, we demonstrate evidence for previously undetected SARS-CoV-2 infections in NYC prior to the first reported case and the emergence of the city as a COVID-19 epicenter in March 2020[4,5].

## Results

To systematically delineate the arrival of SARS-CoV-2 in NYC, we secured 3040 residual nasopharyngeal swab specimens collected in viral transport medium that were banked from patients with respiratory symptoms or influenza-like illness who presented to the MSHS during the first 10 weeks of 2020 (epidemiological weeks ending on 4 January to 7 March), but were found negative by diagnostic molecular amplification testing for routine respiratory pathogens. The number of these residual respiratory pathogen-negative (RPN) specimens collected at each MSHS site varied among the MSHS hospitals as well as from week to week (Supplementary Fig. 1a).

To increase our screening capacity and ensure specimen de-identification, we combined equal volumes of viral transport media from ten distinct RPN specimens into single tubes, yielding 304 pools that underwent nucleic acid amplification testing (NAAT) for SARS-CoV-2 (Fig. 1a) using the Roche Diagnostics cobas® 6800 SARS-CoV-2 Test. This assay, which has emergency use authorization from the US Food and Drug Administration (FDA) for the detection of SARS-CoV-2 in clinical specimens, evaluates samples for the presence of two viral targets: the SARS-CoV-2-specific ORF1ab gene and the pan-Sarbecovirus envelope E gene. Of the 304 RPN pools, both ORF1ab and E gene were detected in eight pools (ORF1ab+E+, 2.6%), only ORF1ab was detected in one pool (ORF1ab+, 0.4%), only E gene was detected in eight pools (E+, 2.6%), and neither was detected in 287 pools (94.4%) (Fig. 1b). Five E+ RPN pools contained specimens from patients treated at two distinct MSHS hospitals (A and C), collected during the weeks ending on 18 January, 25 January, and 1 February (Fig. 1c). None of the RPN pools comprised of specimens collected during the

following 3 weeks yielded detectable SARS-CoV-2 RNA. However, for specimens collected in the week ending on 29 February, SARS-CoV-2 RNA was detected in 5.4% of RPN pools ($n = 3$: 1 ORF1ab+E+, 1 ORF1ab+, and 1 E+); this percentage increased to 33.3% ($n = 9$: 7 ORF1ab+E+, 1 ORF1ab+, and 1 E+) for RPN pools from the week ending on 7 March. These data indicate that SARS-CoV-2 infections were present in a small number of patients seeking care at MSHS facilities across NYC several weeks prior to the first pandemic wave. The high number of positive RPN pools in the first week of March provides an explanation for the sudden exponential increase in severe COVID-19 cases that were admitted to MSHS hospitals starting mid-March 2020.

**Sequencing SARS-CoV-2 in RPN pools.** To validate the NAAT results and to reconstruct the SARS-CoV-2 genomes in these pooled RPN specimens, we extracted viral RNA from all RPN pools with detectable SARS-CoV-2 RNA and performed viral genome sequencing as described previously[4]. We obtained complete SARS-CoV-2 genomes with distinct genotypes from six of the eight ORF1ab+E+ pools (Fig. 2a). To assess for the presence of more than one distinct viral genome in these pools, we determined the fraction of non-consensus viral variants for all positions in each assembly. The maximum fraction of non-consensus variants at any position did not exceed 20%, suggesting that each pool was dominated by a single viral variant.

Eleven pools with detectable SARS-CoV-2 RNA (2 ORF1ab+E+, 1 ORF1ab+, and 8 E+) yielded either scattered or no SARS-CoV-2 reads, suggesting that viral RNA levels in these pools were insufficient to obtain complete genomes. Indeed, these pools had high Ct values for both targets by NAAT assay (e.g., ≥34.25 for ORF1ab, ≥35.63 for E) (Supplementary Fig. 1b). To improve sequence recovery of limited or degraded viral RNA, we repeated sequencing with our custom protocol and a different protocol with smaller tiling amplicons targeting the full genome and regions with clade-defining mutations (see "Methods"). The additional sequencing data allowed us to complete another SARS-CoV-2 genome from ORF1ab+E+ pool P58 (Fig. 2a). The remaining two pools with ORF1ab detected yielded partial genomes (35% genome completeness for ORF1ab+ pool P34, and 24% for ORF1ab+E+ pool P51) (Fig. 2b). We were not able to assemble consensus genome sequences from any of the E+ pools, but three pools from weeks ending on 25 January (P275) and 1 February (P263 and P271) each yielded scattered SARS-CoV-2 reads throughout the viral genome (Fig. 2b and Supplementary Table 1), confirming the presence of viral genetic material. Clade-defining sites were not sufficiently covered to assign these pools to specific clades or lineages.

**Phylogenetics of SARS-CoV-2 from RPN pools.** We next reconstructed phylogenetic relationships between each of the seven early complete genomes (≥95% genome coverage) and a representative dataset of available genomes from the US as well as from viruses circulating globally between January and March 2020 (Fig. 2c and Supplementary Fig. 2). In order to place these genomes on a timed tree reconstruction, we conservatively used the week ending date of each pool. All RPN SARS-CoV-2 genomes were identified in specimens collected in the last week of February (ending on 29 February) and the first week of March (ending on 7 March), a time period when molecular diagnostic testing still was limited to individuals fulfilling a very narrow range of testing criteria. The sequences from these early infections map to four different PANGO lineages, B.1 ($n = 4$), B.1.5 ($n = 1$), B.2.12 ($n = 1$), and A.2 ($n = 1$), consistent with multiple independent introductions (Fig. 2c, Supplementary Fig. 2, and Table 1). All four lineages were detected subsequently during the

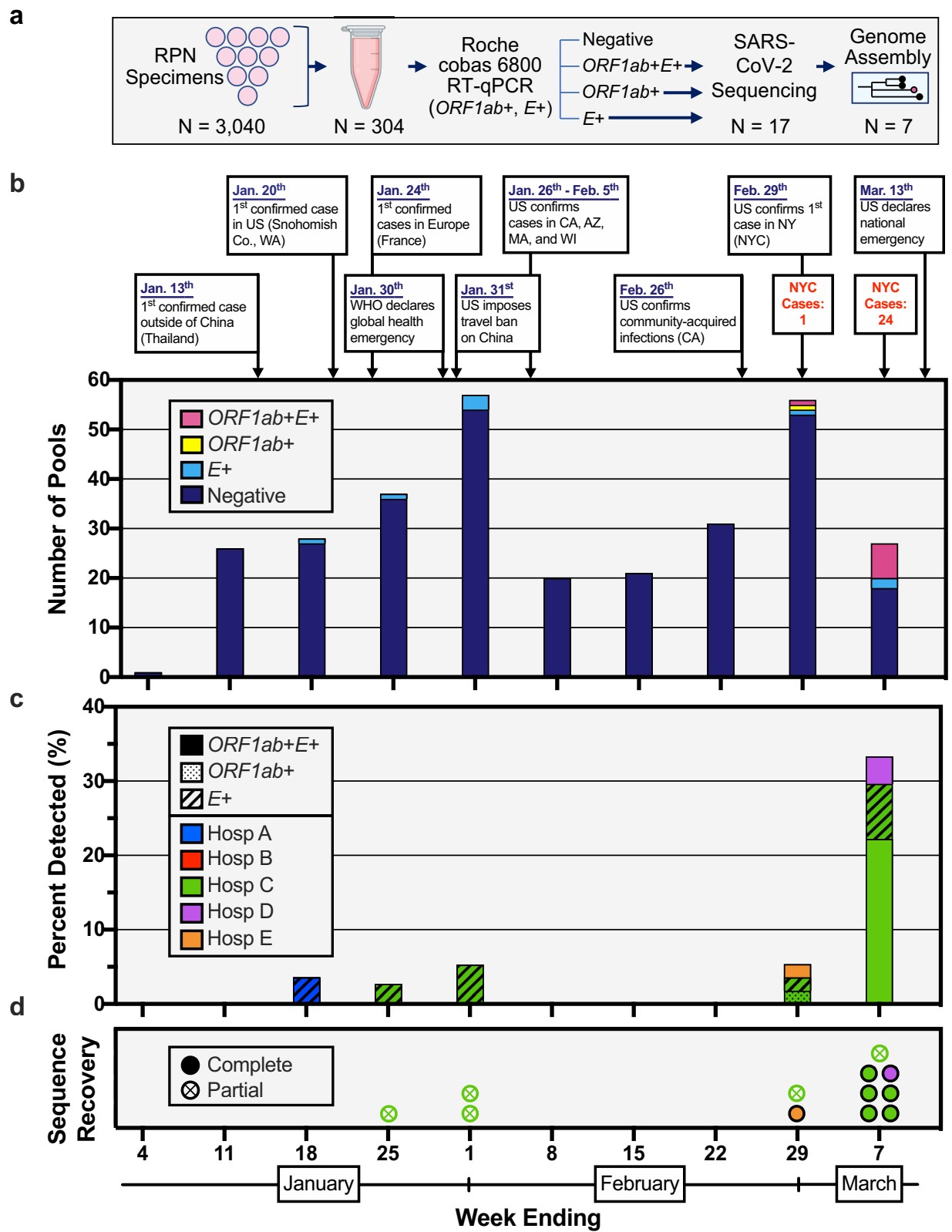

peak in the spring of 2020 in NYC, which was dominated by the B.1 lineage[4,5]. Notably, the B.1 RPN pools (P134, P41, P58, and P53) are nested within a cluster that was linked to the early community spread of SARS-CoV-2 in NYC, delineated by the additional substitutions ORF3a-Q57H and ORF1a-T265I[4,5].

Global SARS-CoV-2 sequencing data between January and March 2020 is scarce, making it difficult to fully resolve the relationships between community spread and additional introductions during the early period of the outbreak in NYC. Lineage B.1 is a parent lineage to multiple emerging lineages that continue to circulate in

**Fig. 1 Detection of SARS-CoV-2 nucleic acids in nasopharyngeal specimens collected in the first 10 weeks of 2020. a** Schematic representation of the study design. Nasopharyngeal swab specimens that tested negative for respiratory pathogens (RPN) were pooled. Each pool consisted of ten specimens from the same week from one of five hospital sites. Nucleic acid amplification testing (NAAT) was performed and RNA was processed for SARS-CoV-2 genome assembly. **b** Select events and responses to the evolving SARS-CoV-2 pandemic are annotated over the timeframe surveyed. Confirmed cases in NYC for the last 2 weeks are noted. Absolute counts of pools that tested positive or negative for RT-PCR targets (ORF1ab+E+ (magenta), ORF1ab+ (yellow), E+ (cyan), Negative (dark purple)) are depicted by week collected. **c** Distribution of pools with RT-PCR target results (ORF1ab+E+ (solid), ORF1ab+ (dotted), E+ (cross-hatched)) across the five different hospital sites in NYC (Hospital A, blue; B, red; C, green; D, purple; E, orange). **d** Distribution of SARS-CoV-2 sequences recovered by collection week and hospital site of RPN pools. Filled points reflect complete SARS-CoV-2 consensus genomes recovered and points with X's reflect partial genomes recovered (e.g., incomplete genomes and those validated by SARS-CoV-2 reads). Colors denote hospital sites as indicated by the legend in (**c**).

the US, including lineage B.1.2 (N:P67S), which has become a dominant lineage in the US (NextStrain build, 17 February 2021, https://nextstrain.org/ncov).

## Discussion

Taken together, we provide clear evidence that SARS-CoV-2 infections were present in NYC at least 6 to 8 weeks prior to the surge of cases that flooded the NYC health system. Previous studies have suggested cryptic transmissions weeks prior to the first confirmed cases of community spread[7,8]. Large retrospective testing efforts have probed for SARS-CoV-2 in banked nasopharyngeal specimens from at least seven states (Michigan, Pennsylvania, Tennessee, Texas, Wisconsin, Washington State[9], and California[10]), with the earliest positive specimens dating back to 21 February 2020 (Seattle, WA[7] and California[10]). In addition, a recent serosurvey of blood products further suggests early undetected spread in multiple states across the US from December 2019 through January 2020[11]. Of note, these studies relied solely on molecular testing without validation by viral genome sequencing. Our study is complementary to those efforts and provides information regarding the presence of SARS-CoV-2 in the diverse, densely populated, international travel hub of NYC, >1 month prior to the detection of the first reported NYS case.

Although we detected SARS-CoV-2 RNA in specimens from late January 2020, without fully reconstructed genomes it is impossible to determine whether these cases seeded the community spread observed in March. Our molecular findings are in agreement with previous evidence of sporadic SARS-CoV-2 infections in the US in January 2020[9], and with evidence from an MSHS SARS-CoV-2 serosurvey[6] that identified low levels of SARS-CoV-2 antibody positivity as early as mid-February 2020, consistent with infection at least 2 weeks earlier. Lastly, although our survey only examined RPN specimens collected starting 30 December 2019, the absence of SARS-CoV-2-positive specimens from early January, in conjunction with the serological evidence[6], makes the presence of SARS-CoV-2 in the US East Coast populace prior to 2020 unlikely.

The observation that the majority of SARS-CoV-2 genomes identified from RPN pools, including specimens collected between the last week of February and the first week of March 2020, cluster within the B.1 lineage is consistent with the phylogenetic analyses by us and others linking most cases during the first wave to an influx of travelers from Europe[4,5,12] prior to travel restrictions on mainland European countries (on 13 March 2020) and the United Kingdom and Ireland (on 16 March 2020). Our findings provide further evidence that the limited availability of diagnostic testing early in the epidemic hindered the identification of SARS-CoV-2-infected individuals[8,9,13,14] and help explain the expansion of the epidemic notwithstanding travel restrictions designed to limit further introductions of SARS-CoV-2 into the US. These observations indicate a brief window

of opportunity in which surveillance, testing, and contact tracing of a limited number of infections may have stemmed community spread.

Our study has several limitations because the RPN pools were made from available residual diagnostic specimens that varied with respect to duration and conditions of storage. It is, therefore, possible that some positive specimens—particularly those with low viral titers—were missed due to degradation of the viral RNA genomes. Of note, our pools of ten specimens each were prepared prior to the US FDA authorization for pooling up to six for diagnostic testing with the cobas® 6800 SARS-CoV-2 real-time RT-PCR Test. Although this pooling strategy represents a slightly larger dilution factor, pooling strategies of 4 up to 30 specimens have been proposed for screening of asymptomatic populations[15]. We started systematically banking RPN specimens in February 2020 and, as a result, may have missed some RPN specimens obtained in early January. However, we included all available residual RPN specimens in our study without any selection. Furthermore, although we reconstructed a single, dominant viral genome from each pool, it is possible that other distinct SARS-CoV-2 variants were present at lower levels. Thus, our estimates regarding the frequency of SARS-CoV-2 positivity over time are a conservative approximation. Lastly, we lack demographic and epidemiological information for individual cases since we relied on de-identified pooled specimens.

An important feature of this study is that we were able to begin retrieval and banking of RPN specimens from our laboratories immediately following the 8 January 2020 CDC Health Alert Network advisory regarding the outbreak of pneumonia of unknown etiology, later identified as COVID-19. Due to space constraints, most clinical laboratories are unable to store negative specimens for more than 1 week. Retrieval of specimens from symptomatic individuals who do not yield routine pathogens can be an important undertaking to survey and identify novel pathogens. Systematic, unbiased surveillance of clinical specimens obtained from individuals presenting with unexplained or unusual clinical presentations of respiratory illness for the presence of emerging viral pathogens must be a key component of any future early-warning sentinel programs. Population-dense metropolitan areas and major global travel hubs present not only a heightened risk for community spread but also an opportunity for monitoring and prevention. These systematic measures will need to become essential components of our new normal in order to prevent local infections and transmissions from blooming into uncontrolled outbreaks.

## Methods

**Ethics statement.** This study was reviewed and approved by the Institutional Review Board of the Icahn School of Medicine at Mount Sinai (ISMMS) (protocol: HS# 20-00141).

**SARS-CoV-2 specimen collection and testing.** RPN specimens were clinical specimens that tested negative for routine testing of respiratory pathogens. These

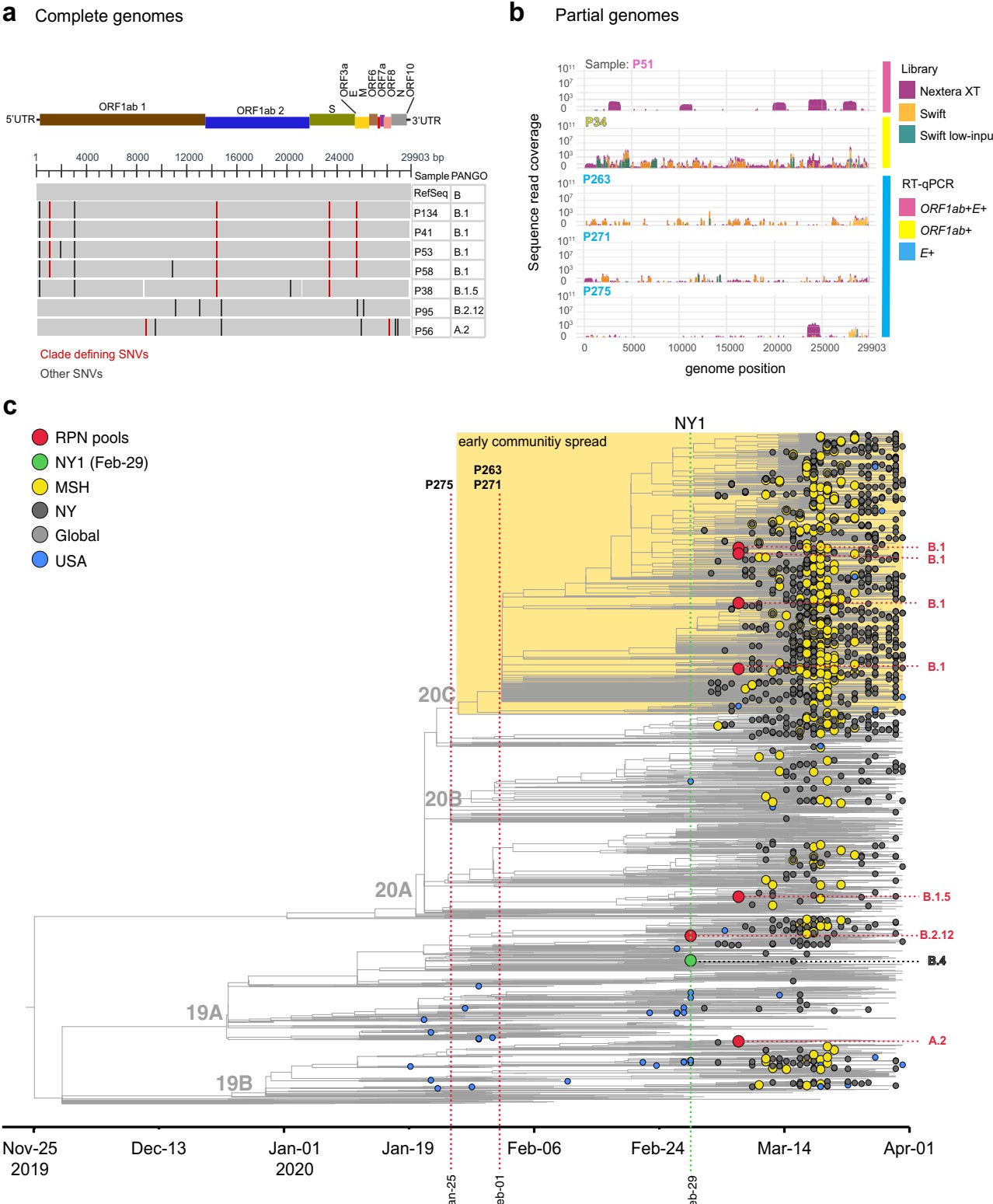

specimens were obtained from inpatients and outpatients and were not stratified by age or other demographic parameters prior to the de-identification process to ensure anonymity. Overall, MSHS Clinical Laboratory data indicate that RPN nasopharyngeal specimens collected during January and February 2020 were obtained during visits to the emergency departments (60%), other outpatient areas (15%), and from inpatients (25%). Most (87%) RPN specimens were from individuals older than 18 years.

The five hospital sites that were sampled for this study are located in the NYC boroughs of Manhattan (*n* = 4) and Brooklyn (*n* = 1). These facilities and associated ambulatory practices serve the greater NYC metropolitan area, including the five NYC boroughs, Westchester County, Long Island, and parts of New Jersey and Connecticut.

RPN specimens were stored at +4 °C for 7 days after clinical testing. These residual RPN specimens were shipped weekly to the centralized clinical microbiology laboratory and frozen at −80 °C until thawed for pooling. RPN pools were aliquoted and stored at −80 °C until testing. Genomes recovered from RPN pools in this study were compared to later sequences obtained from individual clinical specimens from cases that tested positive for SARS-CoV-2 in MSHS once testing became more widely available. Details on testing using the aforementioned systems were previously described[4].

**Fig. 2 Phylogenetic relationships of previously undetected SARS-CoV-2 and other NY and global isolates. a** Multiple sequence alignment of >95% complete SARS-CoV-2 genome sequences obtained from RPN pools relative to Wuhan-Hu-1 (RefSeq: NC_045512). RPN pools are ordered by date and PANGO lineage as displayed in (**a**). The SARS-CoV-2 genome coordinates and gene annotations are shown above. Single- nucleotide variations (SNVs) are depicted with vertical lines in red (clade defining) or black (other). **b** Coverage for pools with detectable RT-PCR targets (*ORF1ab+E+* (magenta), *ORF1ab+* (yellow), *E+* (cyan)) collected prior to the first confirmed case in NY (NY1) with detectable SARS-CoV-2 reads that could not be assembled to complete genomes (>Q30 reads are shown). Nextera XT comprises data from both whole-genome and targeted amplicon sequencing library preparations. **c** Maximum-likelihood (ML) phylogenetic inference shown as a time tree of seven SARS-CoV-2 genome sequences from this surveillance study in a global background of 2993. Tip circles indicate the position of the respiratory pathogen-negative (RPN) pools (red) described in this report, the first reported COVID-19 case in NYC (green) from 29 February, later NYC cases from MSHS (yellow) and other institutions (dark gray), and US (blue) early isolates prior to 1 March. Tips without circles correspond to the background global isolates. The yellow box delineates the position of the clade containing the majority of NYC sequences detected during the early spread. The PANGO lineage classification of the RPN pools is indicated on the right, and the NextStrain clades are shown as node labels. The specimen identifier is indicated for RPN pools detected earlier than NY1. The time tree was inferred under a strict clock model with a nucleotide substitution rate of $0.80 \times 10^{-3}$.

| Table 1 Lineage classification of previously undetected SARS-CoV-2 in NYC. | | | | | | | |
|---|---|---|---|---|---|---|---|
| Sample | Week ending | Ct values (*ORF1ab, E*) | Genome completeness (%) | NextStrain clade | Clade-defining mutations | PANGO lineage | Lineage detection prior to 1 March |
| Pool-134 | 7-Mar-2020 | 18.62, 18.63 | 99.9 | 20C | S:D614G, ORF1b: P314L, OFR3a: Q57H, ORF1a: T265I | B.1 | Mainly Europe, linked to Italian outbreak, only a few North American (non-US) isolates |
| Pool-41 | 7-Mar-2020 | 28.94, 29.36 | 99.7 | 20C | S:D614G, ORF1b: P314L, OFR3a: Q57H, ORF1a: T265I | B.1 | |
| Pool-58 | 7-Mar-2020 | 31.66, 33.65 | 99.8 | 20C | S:D614G, ORF1b: P314L, OFR3a: Q57H, ORF1a: T265I | B.1 | |
| Pool-53 | 7-Mar-2020 | 20.64, 21.07 | 99.8 | 20C | S:D614G, ORF1b: P314L, OFR3a: Q57H, ORF1a: T265I | B.1 | |
| Pool-38 | 7-Mar-2020 | 28.91, 29.43 | 98.9 | 20A | S:D614G, ORF1b: P314L | B.1.5 | Europe/South America/ Asia |
| Pool-95 | 29-Feb-2020 | 20.80, 21.11 | 99.8 | 19A | T14408C | B.2.12 | Asia/Europe/Oceania |
| Pool-56 | 7-Mar-2020 | 25.37, 25.65 | 99.7 | 19B | C8782T ORF8:L84S | A.2 | Europe |

**Preparation of RPN pools**. RPN pools were prepared by mixing aliquots from nasopharyngeal specimens in viral transport medium from patients with respiratory symptoms that previously tested negative for routine respiratory pathogens using multiplex diagnostic panels (e.g., BioMerieux FilmArray Respiratory Panel, Cepheid Xpert® Xpress Flu/RSV). RPN specimens ($n = 3040$) collected at MSHS hospitals and outpatient facilities between 30 December 2019 and 7 March 2020 were organized into groups of ten, and stored at −80 °C. Notably, these specimens had not previously been tested for SARS-CoV-2. The RPN pools ($n = 304$) were prepared in an isolated class II biological safety cabinet at a separate location from the Clinical Microbiology and research labs that had never been used for handling respiratory specimens or viruses.

Briefly, 400 μL of viral transport medium from each specimen was manually aliquoted one at a time into a sterile 5 mL snap-cap centrifuge tube (ASi, C2520). Once each specimen was aliquoted, the 4 mL volume was mixed manually by pipetting and 600 μL aliquots were reserved for SARS-CoV-2 NAAT. RPN specimens and pools were stored at −80 °C.

**SARS-CoV-2 NAAT**. To test for SARS-CoV-2 in RPN pools, 600 μL aliquots underwent NAAT by the cobas® 6800 SARS-CoV-2 real-time RT-PCR Test (Roche, 09175431190) in the MSHS Clinical Microbiology Laboratory, which is certified under Clinical Laboratory Improvement Amendments of 1988 (CLIA), 42 U.S.C. §263a and meets the requirements to perform high complexity tests. Aliquots were run in batches with one cobas® Buffer Negative Control (BUF (−) C) (Roche, 07002238190) and one cobas® Positive Control (SARS-CoV-2 (+) C) (Roche, 09175440190). The assay utilizes two targets to detect SARS-CoV-2 RNA: the SARS-CoV-2-specific *ORF1ab* gene and the pan-*Sarbecovirus* envelope *E* gene. All target results were valid across all 304 RPN pools tested.

**Optimized extraction of total RNA from pools**. Total RNA was extracted manually from 1 mL aliquots of each RPN pool positive for *ORF1ab* and/or *E* gene targets by reverse transcription-polymerase chain reaction (RT-PCR), utilizing the QIAamp UltraSens Virus Kit (Qiagen, 53706) and using an optimized protocol. Prior to extraction, all RPN pools were equilibrated to room temperature for at least 30 min. Briefly, 1 mL of the pooled viral transport medium was transferred to a 2 mL Dolphin Tube (Genesee Scientific, 24-284), lysed by adding 800 μL Buffer AC, and manually mixed by pipetting up and down. Carrier RNA (5.6 μL) was added to each tube and each mixture was vortexed one at a time. Lysates were incubated at room temperature for 10 min and spun at $5000 \times g$ for 3 min. Tubes were opened and supernatants were removed from each tube within a biological safety cabinet.

Lysates were moved to an isolated clean research space designated for nucleic acid extraction. A mixture of Buffer AR (300 μL) and proteinase K (20 μL) prewarmed to 60 °C was added to each lysate, which was then vortexed for 20 s. Lysates were incubated on a ThermoMixer C (Eppendorf, 2231000667) at 40 °C, shaking at 2200 r.p.m. for 10 min. Lysates were then spun down and 300 μL Buffer AB was added to each tube. Mixtures were vortexed for 10 s and RNA was purified by manual extraction on QIAamp spin columns and eluted in 50 μL of AE Elution Buffer for downstream confirmatory RT-PCR testing and sequencing applications.

**SARS-CoV-2 whole-genome amplification and sequencing**. SARS-CoV-2 sequencing was performed on the Illumina MiSeq platform following ProtoScript II (New England Biolabs, E6560) complementary DNA synthesis with random hexamers, SARS-CoV-2 whole-genome amplification with custom-designed tiling primers (Supplementary Table 2)[4], and library preparation with the Nextera XT DNA Sample Preparation Kit (Illumina, FC-131-1096).

For each pool that did not yield a complete genome in the initial sequencing attempt, four additional sequencing libraries were prepared from re-extracted RNA. (1) Nextera XT Illumina amplicon sequencing as described above, (2) Nextera XT sequencing of 1.5–2 kb amplicons targeting only regions containing clade-defining single-nucleotide variation (SNVs) (positions 1059, 8782, 14,408, 23,403, 25,563, 28,144, 28,881, and 28,882, https://nextstrain.org/blog/2020-06-02-SARSCoV2-clade-naming), and the Swift Normalase® Amplicon Panel SARS-CoV-2 (Swift Bioscience COVG1V2-96, SN-5×296 and SN-5S1A96) according to the manufacturer's instructions for (3) regular input and (4) low input samples. Data from the two Nextera XT libraries were combined for assembly shown as "Nextera XT" on Fig. 2b.

**SARS-CoV-2 genome assembly**. Illumina data were analyzed using a custom reference-based (MN908947.3) pipeline, https://github.com/mjsull/COVID_pipe (mjsull 2020)[16], to reconstruct SARS-CoV-2 genomes.

**SARS-CoV-2 phylogenetic analysis and lineage assignment**. Phylogenetic relationships of the seven high-quality consensus sequences (>80% completeness) were inferred over a global background of SARS-CoV-2 sequences between December 2019 and April 2020 downloaded from Global Initiative on Sharing Avian Influenza Data (GISAID) with a few modifications. For the background set, only sequences with >95% non-ambiguous sites were included, and sequences were masked at the 5′ and 3′ ends to remove ambiguous regions but conserve untranslated regions that contained SNVs across the whole dataset. Initial alignment and subsampling were done by using the NextStrain tool (v1.0)[17]. For cases with available information on epidemiological links, or patients with longitudinal sampling when known, only one representative sequence was kept. A maximum-likelihood (ML) phylogeny was inferred with IQ-TREE under the GTR + F + I + G4 model[18,19], after which further manual curation was done to identify and remove extreme outliers that deviated from a temporal signal using Tempest[20]. The final ML tree containing 3700 taxa was then time-scaled with TreeTime using a skyline coalescent tree prior and a strict clock model. The analysis was run for six iterations to improve optimization and resolution.

Lineage classification was done using a phylogenetic-based nomenclature as described by Rambaut et al.[21] using the PANGOLIN tool, lineages version 2020-10-03[22].

**Display items**. All figures are original and were generated using the GraphPad Prism software 9.1.0, R software package ggplot2, Figtree v.1.4.4[23], NCBI Multiple Sequence Alignment Viewer v.1.17.0 (https://www.ncbi.nlm.nih.gov/tools/msaviewer/), BioRender.com, and finished in Adobe Illustrator 2021 (v.25.2.1).

**Reporting summary**. Further information on research design is available in the Nature Research Reporting Summary linked to this article.

## Data availability
SARS-CoV-2 sequencing read data for all study isolates and sample pools were deposited in SRA. BioProjectID PRJNA717974 and BioSample accessions SAMN18520300 to SAMN18520311.

## Code availability
Illumina data were analyzed using a custom reference-based (MN908947.3) pipeline, https://github.com/mjsull/COVID_pipe[16].

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

## Acknowledgements
We thank the members of the Simon and van Bakel laboratories for pitching in whenever additional help was needed. We also acknowledge the assistance provided by the Rapid Response Laboratories and Clinical Microbiology Laboratory of the Mount Sinai Health System with regard to the banking and transfer of nasopharyngeal specimens. We also thank Denis Ruchnewitz and Michael Lässig for their input on the phylogenetic analyses as well as Catherine Teo for her efforts in RT-qPCR assays. We are grateful for the continuous expert guidance provided by the ISMMS Program for the Protection of Human Subjects (PPHS). We also acknowledge the authors and the originating and submitting laboratories of sequences from GISAID's EpiFlu and EpiCoV (www.gisaid.org) that were used as background for our phylogenetic inferences. The Research reported in this paper was supported by the National Institutes of Health (NIH) contract number HHSN272201400008C, the NIH Office of Research Infrastructure under award numbers S10OD018522 and S10OD026880, and institutional and philanthropic funds (Open Philanthropy Project, #2020-215611), as well as a Robin Chemers Neustein Postdoctoral Fellowship Award (to Dr. Gonzalez-Reiche).

## Author contributions
M.M.H., S.F., J.T., A.E.P.-M., M.R.G., M.D.N., and E.M.S. provided clinical samples for the study. M.M.H., S.F., J.T., A.E.P.-M., M.R.G., M.D.N., H.v.b., V.S., and E.M.S. accessioned clinical samples. M.M.H., H.A., C.-Y.W., and A.E.P.-M. performed RNA extraction and measured viral titers. A.S.G.-R., Z.K., A.v.D.G., B.A., and H.v.B. performed NGS experiments. J.S., S.H.S., Y.-C.W., M.S., E.E., and R.S. provided NGS services. A.O. and M.J.S. developed an assembly pipeline and performed genome assembly. M.M.H., A.S.G.-R., A.O., M.J.S., C.C.-C., M.L., F.K., H.v.b., V.S., and E.M.S. analyzed, interpreted, or discussed data. M.M.H., A.S.G.-R., H.v.b., V.S., and E.M.S. wrote the manuscript. H.v.b., V.S., and E.M.S. conceived the study. H.v.b., V.S., and E.M.S. supervised the study. H.v.b., V.S., and E.M.S. raised financial support.

## Competing interests
R.S. is VP of Technology Development and a stockholder at Sema4, a Mount Sinai Venture. This work, however, was conducted solely at Icahn School of Medicine at Mount Sinai. Otherwise, the authors declare no competing interests.
