## [Peer Review File · Nature Communications]

REVIEWER COMMENTS

Reviewer #1 (Remarks to the Author):

The manuscript describes the retrospective testing of stored respiratory samples from more than 3,000 symptomatic patients in the NYC area for SARS-CoV-2, during the first 10 weeks of 2020. Using a pooled testing strategy, the virus was detected in samples from as early as mid-January 2020. Whole genome sequence analysis demonstrated multiple lineages even in these early weeks of introduction of the virus.

The work demonstrates what has been suggested in earlier publications, that SARS-CoV-2 was circulating in NYC before the first previously documented case. Further, it reiterates that if testing had been performed more extensively at an earlier time, cases may have been detected, contact tracing implemented earlier, and the extent and size of the subsequent pandemic wave better controlled.

Some additional information would be helpful to clarify some points in the paper.

1. Were the samples from inpatients, outpatients or a mixture of both? Were they pediatric or adults? While the testing was deidentified, were any patient parameters such as these noted before the deidentification process?

2. The paper mentions multiple facilities in the network. Across what geographic area does the network serve?

3. For the initial testing, pools of 10 samples were tested on a Roche 6800. The FDA approved method for testing pooled samples on this instrument is with a maximum pool size of 6 specimens. Pools of 10 would have decreased the sensitivity of detection. While pools of 6 would have clearly increased the number of tests, it is disappointing to compromise the assay. Moreover, the sequencing studies were performed on pooled samples rather than deconvoluting the pools. While this would have also been a more complicated study design given the logistics of multiple sites, it would have been possible to maintain deidentification and it would have afforded better overall success with the sequencing data.

4. The authors note in the discussion section that specimens "varied with respect to duration and conditions of storage." It would be helpful to have a sentence on the details of this in the methods section.

5. It may also be worth commenting in the discussion section that, due to space constraints, the majority of diagnostic laboratories discard specimens within a few days of testing. Therefore, there are limited opportunities to perform these kinds of studies.

Reviewer #2 (Remarks to the Author):

This is a well-written report providing molecular evidence of previously undetected SARS-CoV-2 infections in New York City >1 month prior to the first reported local case on 29 February. In total 3041 patients with respiratory symptoms but negative for routine pathogens, during the first 10 weeks of 2020 were tested by SARS-CoV-2 RT-qPCR utilizing a pooling strategy, followed by SARS-CoV-2 whole genome sequencing of the positive samples (n=17), resulting in 7 whole genomes. The sampling dates of the latter 7 was however on/after Feb 29.

The results are fairly convincing: both the wet lab procedures as well as the bioinformatic analyses including the time tree methodology are sound. The SARS-CoV-2 variants detected nicely match the SNVs detected in the early European isolates. Furthermore, the findings are biologically plausible, and in line with timewise characteristics: in the phase prior to detection of the first local case, the materials for testing were limited and the case definition was more narrow.

The findings of the study are of interest, not only to researchers and statistical data modelling experts in the USA but also beyond the country of investigation, since this situation is very likely to be the case in a wide range of regions all over the world.

One major point:

in the last sentence of the abstract: in fig S1 B it is made clear that the PCR positives had very high Ct value >35 (values that in some settings are considered borderline or even false-positives) and for one of the two PCR targets. This is the data the authors rely on when claiming that there is evidence of infection one month earlier than the first officially reported New York case (Feb 27), since the WGS data are from 7 samples from from Feb 27 and March, and the authors very nicely label this PCR detection as "presumptive" in Fig 1C, which gives a nice perspective (nuance). So, it would be good and cautious to add this nuance to the formulation in the last sentence of the abstract. This is just to inform the reader upfront that the conclusion was drawn based on PCR signals (>ct 35) and the WGS data are from later dates. (It is not surprising that WGS did not result in high genome coverage from these samples, but that is not the point I am trying to raise here.)

Just a few minor remarks/requests:

-Table 1: can the Cycle thresholds be added to this column? Same accounts for Table S1 (here of even more importance).

-Figure 1C: can the 7 WGS positive samples be depicted in this figure? They are included in the green bars at Feb 29th and March 7th, but than it becomes more clear that there are no WGS data available from before Feb 29, the date of the first reported case in New York city.

-In the method section, four additional library prep methods are described. In Table S1 and Figure 2B, only up to three of those are depicted, why not show the rest of these results here?

-Fig 2B: are these selected examples? This may be described in the figure legend, the reader may be puzzling and comparing the six patient sample numbers in this figure with the seven numbers in figure A.

Reviewer #3 (Remarks to the Author):

This is a short report on identifying SARS-CoV-2 positive samples (and generating a few genomes from them), building on previous work from the same authors. Here they used banked samples from New York dating the first quarter of 2020, when there were few (known) SARS-CoV-2 cases in the US and many obstacles to testing. While this article is not immediately relevant to current events, I think it is important, to not only clarify events in the lead up to the first pandemic wave in New York, but also as a contribution to the wider conversation about how local epidemics are established and for feeding into future sentinel surveillance programs. The article is well written and the results look solid to me. I have a few small comments that should be straightforward to address. The most important comment is about data deposition and does not actually require any changes to the manuscript.

Major comments

1. The authors state that consensus sequences have been uploaded to GISAID. This is a problem, because if a pool contained >1 positive samples then it is possible that the consensus genotype never circulated in the community, and could look like a recombinant. That's unlikely from the samples considered here (based on the prevalence and genetic diversity at the time), but if uploading consensus sequences from pooled samples becomes commonplace and is done with more recent samples the chance of depositing artifactual recombinants would be much higher. (The nightmare scenario of course being someone downloading such sequences and inferring recombinants among VOCs). Therefore I think it best to avoid establishing a precedent of uploading consensus sequences generated from pooled samples to GISAID.

Please note that I'm not suggesting the authors make any changes to their analyses or text, beyond perhaps adding an explanatory note to the "Data and Materials" section. It is simply very important that the genomes are deposited in such a way that it is 100% unambiguous that they are derived from pooled samples and cannot be linked to a single patient. Reads can be uploaded to ENA and the sequences can be uploaded to Genbank (with a metagenome tag), but I don't think it's necessary to upload these genomes to GISAID. If the authors insist, then the host should be set to "pooled" and not to "human" and an extra explanatory note should be added.

2. I think the original maximum-likelihood tree should be shown, as a supplementary figure, or even as a main-text figure. The time-scaled tree shown in Fig 2C has the same topology, but since it was scaled (using a fixed clock rate) it has lost some of the information in the branch lengths. A few sentences on the limitations of the phylogenetic inference would also be a useful addition (i.e. no measures of uncertainty for node heights are shown, and there is no discussion on phylogenetic uncertainty).

3. I would appreciate some discussion on the possibility (or impossibility) of the presumptive positive samples being false-positives.

Minor comments

1. Line 101-103: Please state the numbers of pools that tested positive and presumed positive, not just the percentages.

2. Table 1: PANGOLIN lineage should be changed to Pango lineage. Pango is the nomenclature system, Pangolin is the software tool that assigns Pango lineages.

3. The caption to Fig 2A is misleading. All genomes shown in this panel were generated from pools containing samples collected in the weeks ending on 29 Feb (P95) or 7 Mar (the rest). This is not earlier than the first confirmed case in NY (29 Feb), as claimed in the caption (although some samples in the 29 Feb pool may date from a few days earlier, it is impossible to assign an earlier date to P95).

4. Fig 2A: What is the unlabelled gray bar on top without any SNVs? If it is the reference it should be labelled.

5. Fig 2C: This is a phylogenetic inference, not a phylodynamic inference. The legend for Global lineages is misleading. Global lineages are not marked by light gray tip circles, but are marked by the absence of tip circles. (No location can be ascribed to the branches themselves, so a branch being coloured light gray does not imply it is not in the US).

6. Line 139-141: It may make Fig 2C more informative if the authors could highlight the cluster linked to early community spread on the tree.

7. I think the authors could make their paper more self-contained by expanding a little on the phylogenetic methods instead of using "as previously described" multiple times.

8. IQ-TREE was used to obtain the ML tree. It should probably be stated in the methods.

9. Line 367: >5% non-ambiguous sites means genomes with up to 95% ambiguous sites were included. Is that a typo? It's very unlikely that such genomes would be accurately placed in the tree.

10. Why is the relaxed clock model mentioned in the methods if no results with this model are ever shown or discussed?

Point by point response for our manuscript entitled “Molecular Evidence of SARS-CoV-2 in New York Before the First Pandemic Wave” (NCOMMS-21-07661-T).

We thank the reviewers for their very helpful and insightful comments, and have revised the manuscript accordingly. A detailed point-by-point response to each of the comments is included below.

Reviewer #1

The manuscript describes the retrospective testing of stored respiratory samples from more than 3,000 symptomatic patients in the NYC area for SARS-CoV-2, during the first 10 weeks of 2020. Using a pooled testing strategy, the virus was detected in samples from as early as mid-January 2020. Whole genome sequence analysis demonstrated multiple lineages even in these early weeks of introduction of the virus.

The work demonstrates what has been suggested in earlier publications, that SARS-CoV-2 was circulating in NYC before the first previously documented case. Further, it reiterates that if testing had been performed more extensively at an earlier time, cases may have been detected, contact tracing implemented earlier, and the extent and size of the subsequent pandemic wave better controlled.

Some additional information would be helpful to clarify some points in the paper.

1. Were the samples from inpatients, outpatients or a mixture of both? Were they pediatric or adults? While the testing was deidentified, were any patient parameters such as these noted before the deidentification process?

Answer: Specimens were obtained from inpatients and outpatients, and were not stratified by age or other demographic parameters prior to the de-identification process to ensure anonymity. Overall, MSHS Clinical Laboratory data indicate that respiratory pathogen-negative nasopharyngeal specimens collected during January and February 2020 were obtained during visits to the emergency departments (60%), other outpatient areas (15%), and from inpatients (25%). Most (87%) respiratory-pathogen negative samples were from individuals older than 18 years.

The manuscript has been updated to include this information in the Methods, lines 266-272.

2. The paper mentions multiple facilities in the network. Across what geographic area does the network serve?

Answer: The five hospital sites that were sampled for this study are located in the New York City boroughs of Manhattan (n=4) and Brooklyn (n=1). These facilities and associated ambulatory practices serve the greater NYC metropolitan area, including the 5 NYC boroughs, Westchester County, Long Island and parts of New Jersey and Connecticut.

The manuscript has been updated to include this information in the Methods, lines 273-276.

3. For the initial testing, pools of 10 samples were tested on a Roche 6800. The FDA approved method for testing pooled samples on this instrument is with a maximum pool size of 6 specimens. Pools of 10 would have decreased the sensitivity of detection. While pools of 6 would have clearly increased the number of tests, it is disappointing to compromise the assay. Moreover, the sequencing studies were performed on pooled samples rather than deconvoluting the pools. While this would have also been a more complicated study design given the logistics of multiple sites, it would have been possible to maintain deidentification and it would have afforded better overall success with the sequencing data.

Answer: We agree with the reviewer that smaller pools of 6 specimens would represent a smaller dilution factor than the 10 specimen pools utilized in this study. Of note, our pools of 10 specimens each were prepared prior to the US Food and Drug Administration (FDA) authorization for pooling up to 6 for diagnostic testing with the cobas® 6800 SARS-CoV-2 real-time RT-PCR Test. Although this pooling strategy represents a slightly larger dilution factor, pooling strategies of 4 up to 30 specimens have been proposed for screening of asymptomatic populations (*Lohse et al., 2020* (ref. 15)).

The manuscript has been updated to include this information in the Discussion lines 234-237.

4. The authors note in the discussion section that specimens "varied with respect to duration and conditions of storage." It would be helpful to have a sentence on the details of this in the methods section.

Answer: Clinical specimens that were negative by routine testing of respiratory pathogens were stored in the testing laboratory at +4°C for 7 days after testing. The residual RPN Specimens were shipped weekly to the centralized clinical microbiology laboratory (CML), and frozen at -80°C until thawed for pooling. RPN pools were aliquoted and stored at -80°C until testing.

The manuscript has been updated to include this information in the Methods lines 277-280.

5. It may also be worth commenting in the discussion section that, due to space constraints, the majority of diagnostic laboratories discard specimens within a few days of testing. Therefore, there are limited opportunities to perform these kinds of studies.

Answer: We agree with the reviewer and now further emphasize the value of our dataset on lines 245-250 of the revised manuscript.

Reviewer #2

This is a well-written report providing molecular evidence of previously undetected SARS-CoV-2 infections in New York City >1 month prior to the first reported local case on 29 February. In total 3041 patients with respiratory symptoms but negative for routine pathogens, during the first 10 weeks of 2020 were tested by SARS-CoV-2 RT-qPCR utilizing a pooling strategy, followed by SARS-CoV-2 whole genome sequencing of the positive samples (n=17), resulting in 7 whole genomes. The sampling dates of the latter 7 was however on/after Feb 29.

The results are fairly convincing: both the wet lab procedures as well as the bioinformatic analyses including the time tree methodology are sound. The SARS-CoV-2 variants detected nicely match the SNVs detected in the early European isolates. Furthermore, the findings are biologically plausible, and in line with timewise characteristics: in the phase prior to detection of the first local case, the materials for testing were limited and the case definition was more narrow.

The findings of the study are of interest, not only to researchers and statistical data modelling experts in the USA but also beyond the country of investigation, since this situation is very likely to be the case in a wide range of regions all over the world.

One major point:

In the last sentence of the abstract: in fig S1 B it is made clear that the PCR positives had very high Ct value >35 (values that in some settings are considered borderline or even false-positives) and for one of the two PCR targets. This is the data the authors rely on when claiming that there is evidence of infection one month earlier than the first officially reported New York case (Feb 27), since the WGS data are from 7 samples from Feb 27 and March, and the authors very nicely label this PCR detection as "presumptive" in Fig 1C, which gives a nice perspective (nuance). So, it would be good and cautious to add this nuance to the formulation in the last sentence of the abstract. This is just to inform the reader upfront that the conclusion was drawn based on PCR signals (>ct 35) and the WGS data are from later dates. (It is not surprising that WGS did not result in high genome coverage from these samples, but that is not the point I am trying to raise here.)

Answer: We agree with the reviewer that the use of the term "presumptive" can be unclear. To eliminate confusion, we now refer to the gene targets detected in each pool (ORF1ab and/or E gene) throughout the text and the revised Figures.

Just a few minor remarks/requests:

-Table 1: can the Cycle thresholds be added to this column? Same accounts for Table S1 (here of even more importance).

Answer: Cycle thresholds for each target for each pool have now been included in Tables 1 and Supplemental Table 1.

-Figure 1C: can the 7 WGS positive samples be depicted in this figure? They are included in the green bars at Feb 29th and March 7th, but than it becomes more clear that there are no WGS data available from before Feb 29, the date of the first reported case in New York city.

Answer: We have included a new panel Fig. 1d to identify - by site and week collected - which pools yielded complete or partial SARS-CoV-2 genomes.

-In the method section, four additional library prep methods are described. In Table S1 and Figure 2B, only up to three of those are depicted, why not show the rest of these results here?

Answer: We had indicated in the methods section for genome assembly that we combined the reads produced by different Nextera XT library preparations, as both used the same amplification strategy with different primer sets. We have now further clarified this in the legend of Fig. 2b and added a footnote to Supplemental Table 1 as follows: "*Data from two Nextera XT library preparations were combined for assembly shown as "Nextera XT" on Fig. 2b*".

-Fig 2B: are these selected examples? This may be described in the figure legend, the reader may be puzzling and comparing the six patient sample numbers in this figure with the seven numbers in figure A.

Answer: Fig. 2a shows the variant profiles for all complete viral genomes that were obtained in our study. Fig. 2b shows all additional sequencing data we were able to obtain in additional library preparations for specimens that did not yield complete genome data in the initial round. The additional data allowed us to complete the genome for P58, which is why it is included in both panels. However, we understand why this may be confusing, and we have therefore now revised Fig. 2 to show only complete genomes in Fig. 2a (including P58) and only partial genomes in Fig. 2b (excluding P58).

Reviewer #3

This is a short report on identifying SARS-CoV-2 positive samples (and generating a few genomes from them), building on previous work from the same authors. Here they used banked samples from New York dating the first quarter of 2020, when there were few (known) SARS-CoV-2 cases in the US and many obstacles to testing. While this article is not immediately relevant to current events, I think it is important, to not only clarify events in the lead up to the first pandemic wave in New York, but also as a contribution to the wider conversation about how local epidemics are established and for feeding into future sentinel surveillance programs. The article is well written and the results look solid to me. I have a few small comments that should be straightforward to address. The most important comment is about data deposition and does not actually require any changes to the manuscript.

Major comments

1. The authors state that consensus sequences have been uploaded to GISAID. This is a problem, because if a pool contained >1 positive samples then it is possible that the consensus genotype never circulated in the community, and could look like a recombinant. That's unlikely from the samples considered here (based on the prevalence and genetic diversity at the time), but if uploading consensus sequences from pooled samples becomes commonplace and is done with more recent samples the chance of depositing artifactual recombinants would be much higher. (The nightmare scenario of course being someone downloading such sequences and inferring recombinants among VOCs). Therefore I think it best to avoid establishing a precedent of uploading consensus sequences generated from pooled samples to GISAID.

Please note that I'm not suggesting the authors make any changes to their analyses or text, beyond perhaps adding an explanatory note to the "Data and Materials" section. It is simply very important that the genomes are deposited in such a way that it is 100% unambiguous that they are derived from pooled samples and cannot be linked to a single patient. Reads can be uploaded to ENA and the sequences can be uploaded to Genbank (with a metagenome tag), but I don't think it's necessary to upload these genomes to GISAID. If the authors insist, then the host should be set to "pooled" and not to "human" and an extra explanatory note should be added.

Answer: We agree with the reviewer, and had, in fact, not deposited the assemblies from the sample pools to GISAID. The stray data reference to GISAID was included by mistake and we apologize for the oversight. To avoid mis-interpretation of the consensus assemblies from each pool and to aid re-analysis of our data, we opted to deposit the raw SARS-CoV-2 data (after removal of host sequences) in the sequence read archive (SRA), and have included the accession numbers in the revised manuscript (lines 390-391).

2. I think the original maximum-likelihood tree should be shown, as a supplementary figure, or even as a main-text figure. The time-scaled tree shown in Fig 2C has the same topology, but since it was scaled (using a fixed clock rate) it has lost some of the information in the branch lengths. A few sentences on the limitations of the phylogenetic inference would also be a useful addition (i.e. no measures of uncertainty for node heights are shown, and there is no discussion on phylogenetic uncertainty).

Answer. We appreciate the suggestion and now include the Maximum Likelihood tree as a supplementary figure (Supplemental Fig. 2) showing bootstrap values above 70%. We also added the following statement regarding the limitations of the phylogenetic analysis on lines 193-195:

“Global SARS-CoV-2 sequencing data between January and March 2020 is scarce, making it difficult to fully resolve the relationships between community spread and additional introductions during the early period of the outbreak in NYC.”

3. I would appreciate some discussion on the possibility (or impossibility) of the presumptive positive samples being false-positives.

Answer: We recognize that our use of the terminology “presumptive positive” was confusing. Per the EUA documentation of the diagnostic assay utilized, specimens for which only the E gene target is detected are resulted as “presumptive positive”, because the target sequence does not distinguish among the Sarbecoviruses, and without detection of the ORF1ab target could also represent SARS-CoV-1 or MERS. To minimize confusion, we now refer to the specific targets detected in each RPN pool in the revised text and figures. With the exception of the RPN pool for the week ending on 18 January 2020, additional SARS-CoV-2 sequences were found in all RPN pools from which the ORF1ab or E gene targets or both were detected, confirming the presence of SARS-CoV-2. The earliest RPN pool for which additional SARS-CoV-2 reads confirmed the presence of SARS-CoV-2 was from the week ending on 25 January 2020.

Minor comments

1. Line 101-103: Please state the numbers of pools that tested positive and presumed positive, not just the percentages.

Answer: The numbers of pools as well as the percentages are now included in lines 92-94 of the revised manuscript. Please note that to minimize confusion, we no longer use the terms positive and presumptive positive, but also have revised the text to indicate which targets were detected for the pools.

2. Table 1: PANGOLIN lineage should be changed to Pango lineage. Pango is the nomenclature system, Pangolin is the software tool that assigns Pango lineages.

Answer: All mentions of PANGOLIN have now been changed to PANGO lineage, where applicable.

3. The caption to Fig 2A is misleading. All genomes shown in this panel were generated from pools containing samples collected in the weeks ending on 29 Feb (P95) or 7 Mar (the rest). This is not earlier than the first confirmed case in NY (29 Feb), as claimed in the caption (although some samples in the 29 Feb pool may date from a few days earlier, it is impossible to assign an earlier date to P95).

Answer: The reviewer is correct, and the figure legend has now been fixed. The statement “specimens collected prior to the first confirmed case in NY (NY1)” applies to most samples shown in Fig. 2b which were collected prior to the first confirmed case in NY. This sentence was moved to the caption for Fig. 2b.

4. Fig 2A: What is the unlabelled gray bar on top without any SNVs? If it is the reference it should be labelled.

Answer: The reviewer is correct. The top gray bar is the reference sequence. We have labeled it as RefSeq in the figure and legend.

5. Fig 2C: This is a phylogenetic inference, not a phylodynamic inference. The legend for Global lineages is misleading. Global lineages are not marked by light gray tip circles, but are marked by the absence of tip circles. (No location can be ascribed to the branches themselves, so a branch being coloured light gray does not imply it is not in the US).

Answer: We thank the reviewer. This has been corrected in the current version of Fig. 2c.

6. Line 139-141: It may make Fig 2C more informative if the authors could highlight the cluster linked to early community spread on the tree.

Answer: We agree and have now highlighted the cluster linked to early spread (Clade 20C / PANGO lineage B.1) with a yellow box.

7. I think the authors could make their paper more self-contained by expanding a little on the phylogenetic methods instead of using "as previously described" multiple times.

Answer: We added additional details on the methodology for phylogenetic inference (lines 371-375). We have, however, maintained the reference to our previous study as it further expands on the rationale for using these methods and the limitations on the inferences they produce.

8. IQ-TREE was used to obtain the ML tree. It should probably be stated in the methods.

Answer: We now mention IQ-TREE by name on line 503-504: *A maximum likelihood (ML) phylogeny was inferred with IQ-TREE under the GTR+F+I+G4 model^{18,19}. (Ref 18 corresponds to the manuscript describing IQ-TREE by Nguyen et al., 2015).*

9. Line 367: >5% non-ambiguous sites means genomes with up to 95% ambiguous sites were included. Is that a typo? It's very unlikely that such genomes would be accurately placed in the tree.

Answer: Thank you for noting this typo; this sentence has now been corrected to ">95%".

10. Why is the relaxed clock model mentioned in the methods if no results with this model are ever shown or discussed?

Answer: Thank you for noting this. We ran both models, which resulted in comparable results (clock rate for relaxed clock was 0.91×10^{-3}). For simplicity, only the strict clock model results were shown. We have now updated this sentence to only mention the strict clock inference.